# Comparative Analysis of the Reproduction Accuracy of Main Methods for Finding the Mandible Position in the Centric Relation Using Digital Research Method. Comparison between Analog-to-Digital and Digital Methods: A Preliminary Report

**DOI:** 10.3390/ijerph17030933

**Published:** 2020-02-03

**Authors:** Andrey Stafeev, Aleksandr Ryakhovsky, Pavel Petrov, Sergey Chikunov, Aleksandr Khizhuk, Marina Bykova, Nataliya Vuraki

**Affiliations:** 1Department of Prosthodontics, Omsk State Medical University, 644099 Omsk, Russia; andre.a.s@mail.ru (A.S.); san4elo-82@mail.ru (A.K.); 2Central Research Institute of Dentistry and Maxillofacial Surgery, 119021 Moscow, Russia; avantis2006@mail.ru; 3Department of Prosthodontics I.M. Sechenov First Moscow State Medical University (Sechenov 1-st MGMU), 119992 Moscow, Russia; 4Department of Prosthodontics, People’s friendship University (RUDN); 119992 Moscow, Russia; bykova_mv@pfur.ru (M.B.); vuraki_nk@pfur.ru (N.V.)

**Keywords:** centric relation, TMJ, digital dentistry

## Abstract

*Purpose*: To compare the aspect of the reproduction accuracy in studied methods of determination of the (CR) of jaws using the digital research methods. The methods used were bilateral manipulation by P.E. Dawson, frontal deprogrammer, leaf gauge, and intraoral device for recording of Gothic arch angle. *Methods*: To determine the reproduction accuracy of the centric relation of jaws, we examined 5 patients with intact dentition in a prosthetic dentistry clinic (first class in Angle’s system). For each method, 20 registrations of the centric jaw relation were carried out by one operator. The breaks between definitions were 30 minutes. A total of 400 CR recording operations were carried out (400 records of CR). In order to study the reproducibility of CR determination methods, 200 recorded mandible positions were analyzed by means of an analog-to-digital method (a macro kit Canon 650D, Canon 60 mm macro IS USM f2.8, Canon macro ring MR-14 EX and the computer program Adobe Photoshop) to assess the first occlusal contact obtained in the CR of jaws, while the other 200 were analyzed by means of a digital method (the computer program Avantis for 3D modeling, Prime as a laboratory 3D scanner (DOF), and Trios as an intraoral scanner (3Shape)) to assess the spatial position of the mandible in the CR. Statistical analysis was carried out using STATISTICA-10. In all statistical analysis procedures, the critical significance level p was assumed to be 0.05. *Results*: In the study of the data by means of the computer program Avanti 3D, the reproducibility of the mandible position in the CR reached 0.119 ± 0.012 mm for frontal deprogrammer, 0.225 ± 0.028, *p* ≤ 0.05 for bilateral manipulation by Dawson P.E., 0.207 ± 0.02, *p* ≤ 0.05 for leaf gauge, and 0.120 ± 0,013, *p* ≤ 0.05 using an intraoral device for recording the Gothic arch angle. The analog-to-digital method showed an identical tendency for reproduction of the mandible position. *Conclusions*: The digital analysis we made using the Avantis 3D program showed, with high confidence, that the maximum reproducibility of the CR position was reached by using our own design frontal deprogrammer and the device for recording Gothic arch angle.

## 1. Introduction

Currently, in the clinical practice of prosthodontists, the possibility of using digital methods in diagnosing, planning, and treating patients has significantly increased. At the same time, we should pay attention to the fact that doctors are less focused on the implementation of the main clinical stages of prosthodontic tooth rehabilitation, which can lead to blunder and subsequent complications after prosthodontic rehabilitation [1]. One of these stages (the basic stage, according to most researchers) is to determine the centric relation (CR) of the jaws [2,3,4]. Failures associated with CR determination have a high degree of probability of leading to occlusal and muscle disfunction and the appearance of pain, not only in the craniomandibular system, but also in the craniocervical area [5,6,7,8,9,10]. Certain difficulties arise due to interpretation of the definition of the CR concept, the indications and contraindications of determining this position as well as methods for its recording, some of which are very contradictory [3]. According to many studies, there are more than thirty interpretations of the CR concept and of the methods used for its determination [11]. The most common is the technique of bilateral manipulation by P.E. Dawson (Dawson P., 2006), CR registration by frontal deprogrammer (Lucia V. O., 1964) and all of its variation types, leaf gauge (Long J. H., 1970), and intraoral device for recording Gothic arch angle (Gysi A., 1908). Dawson’s technique of bilateral manipulation is manually controlled in the direction of the mandible and following localization of the TMJ condyles in their physiological terminal position. This position is then registered by a special hard wax with addition of aluminum particles (AluWax (Maarc)) for CR registration. The frontal deprogrammer is a bite plate with flat surface that allows the masticatory group of teeth to be divided in order to convert the lower jaw movements in the transversal plane, as condyles are in their terminal position in centric relation.

The leaf gauge is a device consisting of 55 sheets of flexible film, 100 µm thick, which allows the consistent separation of the chewing group of teeth to eliminate proprioceptive sensitivity until the TMJ condyles take their place in the mandible CR. Recording the Gothic arch angle intraorally uses a special device, which includes two parts: the upper, which is the recording site, and the lower with the recording pin, both of which are fixed to the teeth by light-curing baseplate resin (Zhermack, Elite LC Tray). The device is placed in the patient’s mouth and the patient asked to make consistently eccentric movements of the mandible. The recording pin forms a Gothic arch angle on the recording site and its top reaches the CR of the jaws. The choice of the centric relation method and the accuracy of its reproduction are determined by a number of criteria that must be taken into account in the clinic. At the present stage, indications and contraindications for choosing the method of finding the mandible position are not sufficiently recognized, therefore it is very difficult to prioritize a particular method [12]. Only by assessing the degree of precision of the various methods, which would allow the development of certain clinical recommendations for their use, can this problem be solved more effectively.

## 2. Methods

To determine the reproduction accuracy of the centric relation of jaws, we examined 5 patients (3 male and 2 female, in age from 25 to 35) with intact dentition (first class in Angle’s system) in the prosthetic dentistry clinic of Omsk State Medical University. The position of the mandible was assessed using the following methods: bilateral manipulation (Dawson P., 2006) [1]; frontal deprogrammer (Lucia V. O., 1964) [13] of our own design; leaf gauge (Long J. H., 1970) [14]; and device for intraoral recording of Gothic arch angle (Gysi A., 1908) by MJR (Nobilium, USA) [15]. Each method was carried out on 20 registrations of the centric jaw relation by one operator. The breaks between definitions were 30 minutes [16]. A total of 400 CR recording operations were carried out (400 records of CR). 

Registration of the CR was carried out with the use of vinylpolysiloxane material (Occlufast Rock, Zhermack). In order to study the reproducibility of CR determination methods, 200 recorded mandible positions were analyzed by means of an analog-to-digital method to assess the first occlusal contact obtained in the CR; the other 200 were analyzed by means of a digital method to assess the spatial position of the mandible in the CR. 

For the analog-to-digital method, a macro kit (Canon 650D, Canon 60 mm macro IS USM f2.8 and Canon macro ring MR-14 EX) was used and the computer program Adobe Photoshop. Pre-made cast models were plastered by transferred position of the upper jaw using the facial arc (SAM Axioquick). The lower jaw was plastered by CR registration in the articulator (SAM 2 PX). After that, using 8 µm articulation foil (Bausch), the first occlusal contact in the CR was identified and recorded using a macro photography kit. Subsequently the mandible was replastered using the other 199 CR records and the same job was done. The macro photography kit on the tripod and the parallelometer table were fixed at an electronic horizontal level of 0 degrees. The plastered models were then placed on the parallelometer table. The focus distance on the camera was fixed at 0.3 mm. Photos were carried out in manual mode. The white balance was set automatically: ISO 100, exposure 125, aperture size 25, RAW image format. This established identical conditions and excludes any errors due to the photography method. In total, 200 photos were taken by one operator. 

Analysis of the photos, which were loaded into Adobe Photoshop, was carried out. Changes in the position of occlusal contacts for each variant of CR registration were calculated in the mesiodistal and vestibular–oral directions (transversal and distal changes in the position of the mandible). The morphometric parameters of the tooth (mm) were changed to pixels, and a grid with a size of one cell of 5 pixels was applied to the photo, corresponding to 19 µm (Figure 1), so the degree of method error equaled 19 µm. A line drawn from the top of the mounds previously marked on the plaster model was taken as the horizon line. The axis of the vestibular–oral (VO) deflection of the contact was set in parallel to this horizon line. The axis of the mesiodistal (MD) deflection of the contact was carried out at an angle of 90 degrees to the VO axis. We took the occlusal contact obtained during the creation of the first plastered model in the articulator as the reference point in each method, and studied the deviation of occlusal contacts obtained in the subsequent plastered models in CR from this point (Figure 2). The analysis and interpretation were also carried out by one operator.

For the digital method, we used the computer program Avantis for 3D modeling (developed by Professor Ryakhovsky A.N., Russia), Prime (DOF) as the laboratory 3D scanner, and Trios (3Shape) as the intraoral scanner. Digital impressions were obtained using the Trios intraoral scanner and scans of the CR records were obtained by the Prime laboratory scanner, which were integrated into Avantis 3D. The positions of the mandible were set (Figure 3) using digital copies of CR records (a total of 200 items). The starting point was the position of the mandible that was obtained at the first registration by each method. Then, sequentially and in turn, the remaining 9 positions of the mandible were superimposed (Figure 4).

For superimposition analysis, the program “shell–shell comparison” program was used which, when activated, renders the color shade scale of the compared skins, where shades of green correspond to matching and a saturated green color corresponds to the maximum match; shades of blue and red correspond to discrepancies between the skins and the color red or blue (and their shades) are chosen depending on the direction of discrepancy (Figure 5). When hovering over any area of the compared elements, we obtain a numerical value corresponding to the discrepancy in this area. In order to estimate the average coincidence and discrepancy of the compared positions on their entire surface, the digital model taken as a reference point was previously supplied with the help of the program tool “Creating a point” for 12 points, and numerical values were obtained in the following areas: 4 points on top of cusps 3.7 of a tooth, 4 points on the central incisors of the mandible, and 4 points on the top of cusps 4.7 of a tooth (Figure 6). The choice of points in these areas allowed us to take into account the change in the position of the mandible in all directions, and after calculating the average value for the selected points we obtained a number (mm), which showed the average total difference of the compared shells, namely, the two selected positions of the mandible (Figure 7). 

A total of 2160 numerical values was obtained in the area of points and 180 numerical values on the average total discrepancy. Statistical analysis was carried out using STATISTICA-10. In all statistical analysis procedures, the critical significance level p was assumed to be 0.05. Verification of normality of distribution was performed using the Shapiro–Wilk criterion and F- Fisher criterion were used to check hypotheses for the equality of the general dispersions. Parametric methods (t-criterion) were used to check statistical hypotheses.

## 3. Results

In the study of the data by means of the computer program Avantis 3D, the reproducibility of the mandible position in the CR reached 0.119 ± 0.012 mm for frontal deprogrammer, 0.225 ± 0.028, *p* ≤ 0.05 for bilateral manipulation by P.E. Dawson, 0.207 ± 0.02, *p* ≤ 0.05 for leaf gauge, and 0.120 ± 0,013, *p* ≤ 0.05 for intraoral device for recording the Gothic arch angle. 

The most stable reproducibility of the mandible position in the CR, relative to the first registration, was determined by the method using a frontal deprogrammer and was 0.119 ± 0.012 mm, which is significantly less than using the methods of bilateral manipulation (0.225 ± 0.028, *p* ≤ 0.05) and leaf gauge (0.207 ± 0.02, *p* ≤ 0.05). A similar mean value of reproduction accuracy was observed during determination of the CR by recording the Gothic arch angle (0.120 ± 0.013, *p* ≤ 0.05), which also has a significantly lower value (*p* ≤ 0.05) compared to other methods of centric relation determination of the jaws (Table 1). 

Analyzing the data by the analog–digital method, the discrepancy along the VO axis was 0.055 ± 0.03 mm and 0.047 ± 0.02 mm for the frontal deprogrammer along the MD axis0. Equally accurate was the method of CR determination using intraoral recording of the Gothic arch angle (discrepancy on the axis of the VO is −0.097 ± 0.06 mm and on the axis of the MD is −0.055 ± 0.03 mm). Discrepancy of the first contact during determination by means of the leaf gauge on the axis of VO and MD was 0.470 ± 0.16 mm and 0.203 ± 0.10 mm respectively. The greatest inaccuracy in the CR reproduction showed in the method of determination using bilateral manipulation by Dawson; the discrepancy along the axis of VO and MD was 0.746 ± 0.061 and 0.181 ± 0.27 mm, respectively. Comparative analysis (Table 2) of the average discrepancy from the first contact showed a significant increase (*p* ≤ 0.05) of this indicator using the methods of CR determination with leaf gauge and bilateral manipulation in comparison with the frontal deprogrammer and device for recording the Gothic arch angle. The analog-to-digital method showed an identical tendency for reproduction of the mandible position. The obtained values of the degree of precision for the CR determination methods showed a correlation between analog-to-digital and digital methods.

## 4. Discussion

Analyzing the results of the study in terms of assessing the precision of determining the centric relation of the jaws using digital technologies, it should be noted that no method showed 100% reproduction accuracy. The maximum discrepancy in the superposition of digital models of the mandible was determined during bilateral manipulation (0.717 mm) and using a leaf gauge (0.568 mm), and in the first case, the difference between the obtained maximum and minimum (0.004 mm) values is the most distinct (Figure 8 and Figure 9).

The high inaccuracy of the method of bilateral manipulation may be due to the peculiarity of the method, specifically, the formation of a certain manual skill, the practical experience of the doctor, and the peculiarities of the muscular component of the dentoalveolar system. The inaccuracy during determination using the leaf gauge is associated with the potential distalization of the mandible. Thus, our analysis, with a high degree of reliability, showed that the maximum reproducibility in CR determination was seen in methods using the frontal deprogrammer (of our own design) and the device for recording the Gothic arch angle. 

By using the analog-to-digital method we identified the first arising occlusal contact in centric relation, wherein the spatial position of the mandible according to the maxilla remains unknown. Identifying the spatial position of the mandible was not possible according to this method because finding the first occlusal contact was carried out in the articulator. The Avantis 3D program (Developed by Professor Ryakhovsky A.N., Russia) showed the most valuable identification of CR determination in reproducibility of the mandible position because the program allows the comparison of the primary position of the mandible in centric relation without displacement, as if it had been carried out in an articulator. All the data from the digital methods have clinical significance because of their precision in millimeter values, which can be used not only for evaluating the method reproducibility, as in the analog-to-digital method, but also for understanding the movements of the mandible and the TMJ by using each of the CR determination methods in a dental office. 

This study showed that the choice of the method of CR determination depends on factors taken into account in clinical work, specifically, the assessment of the stomatognathic system, the degree of manual skills of the doctor, the psychoemotional state of the patient, and the level of financial security of the clinic, so clinical recommendations need to be studied further. Positions need to be further developed with regard to the functional relationship of the craniomandibular system with the muscle status and TMJ, using modern and objective methods. 

## 5. Conclusions

Our study has shown that all methods for searching for centric relation do not 100% coincide. However, the most precise were methods using the frontal deprogrammer (of our own design) and the intraoral recording of gothic angle. In terms of reproducibility, the Avantis 3D program best identified the mandible position in searching for the position of centric relation. In choosing a method, there are many factors that should be considered in clinical practice, such as the condition of the stomatognathic system, the manual skills of the doctor, the psychoemotional status of the patient, and the material provision of the clinic.

## Figures and Tables

**Figure 1 ijerph-17-00933-f001:**
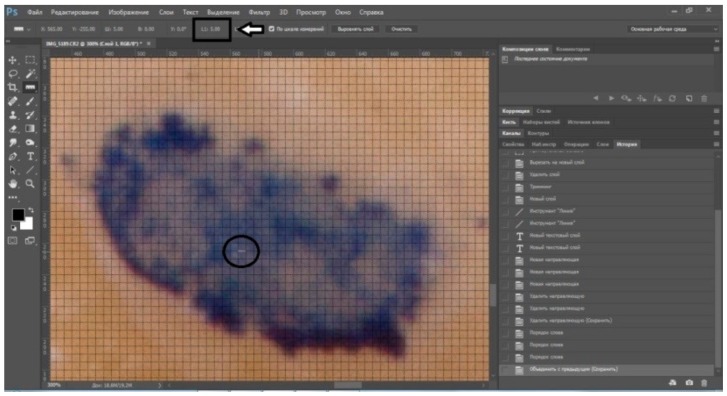
Occlusal contact with a digital grid.

**Figure 2 ijerph-17-00933-f002:**
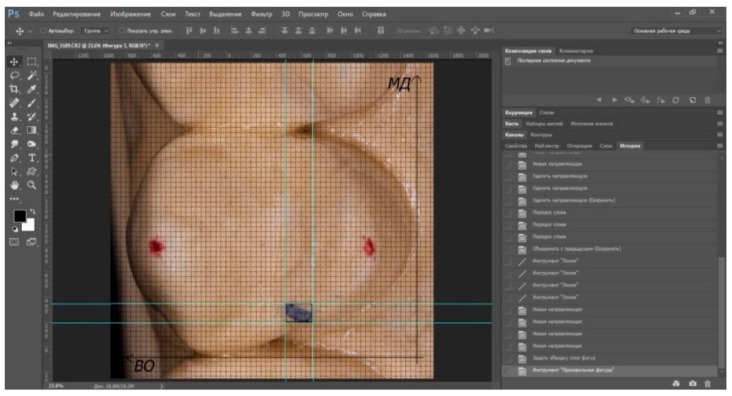
Analysis of deviation of occlusal contact in Adobe Photoshop.

**Figure 3 ijerph-17-00933-f003:**
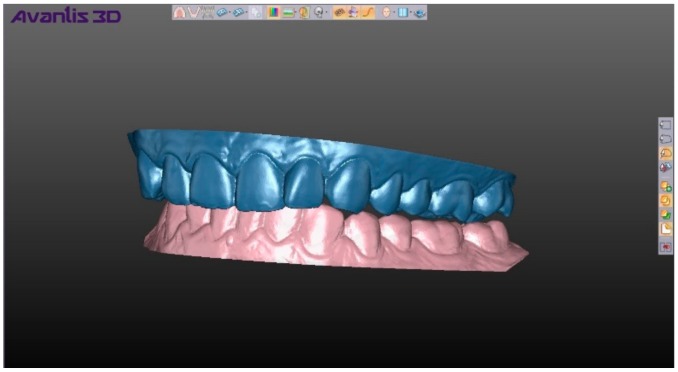
Position of centric relation of the jaws in Avantis 3D program.

**Figure 4 ijerph-17-00933-f004:**
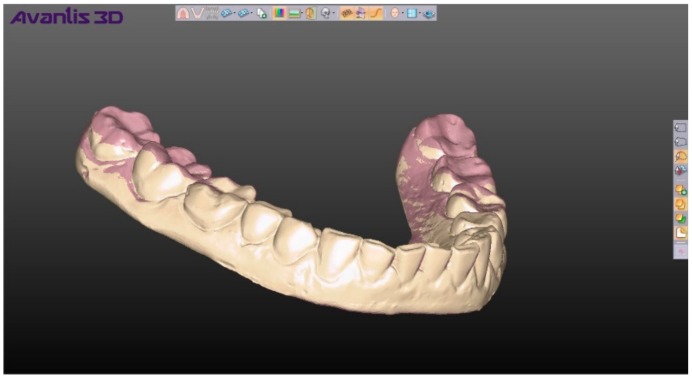
Overlap of two positions (yellow and pink colors) of lower jaw.

**Figure 5 ijerph-17-00933-f005:**
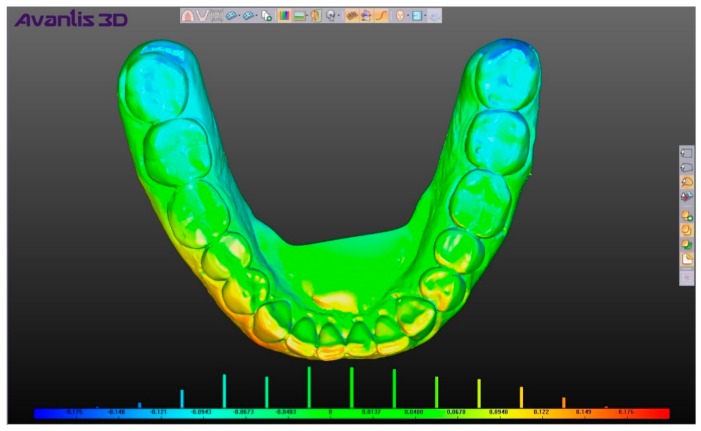
Chart in tool “shell–shell comparison” of program Avantis 3D (range of scale from 0.175 to −0.175 mm).

**Figure 6 ijerph-17-00933-f006:**
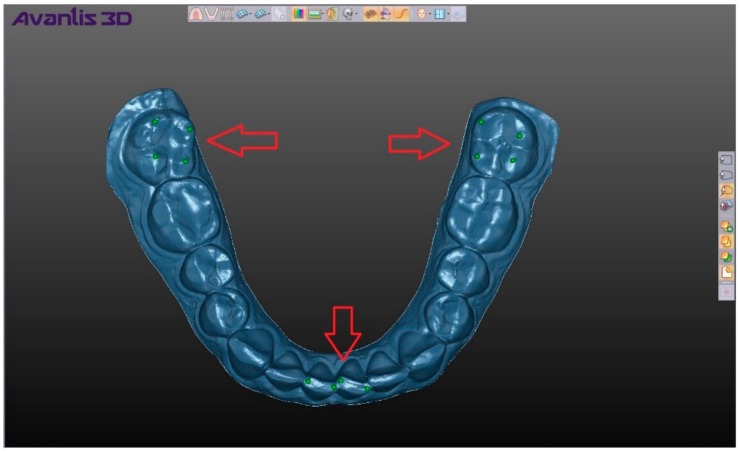
Places for received numeric value of divergence (noted with green point).

**Figure 7 ijerph-17-00933-f007:**
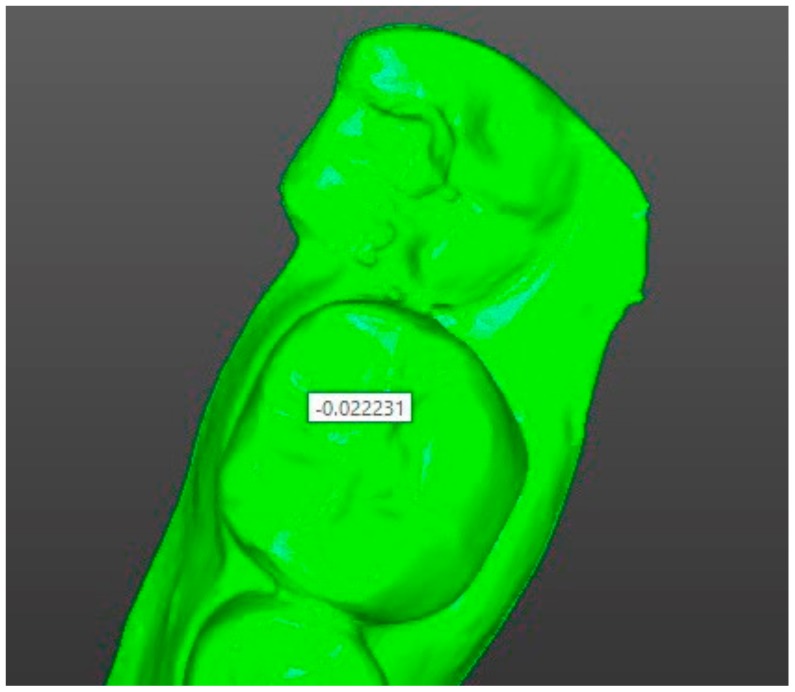
Numeric value of divergence in a chosen area.

**Figure 8 ijerph-17-00933-f008:**
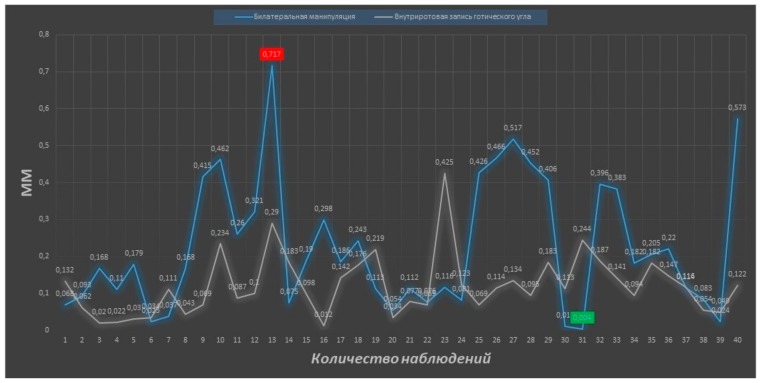
Figure of reproducibility graphic of bilateral manipulation (blue color) and device for recording the gothic angle (grey color). Number of observations at the x-axis, value of deviation in mm at the y-axis.

**Figure 9 ijerph-17-00933-f009:**
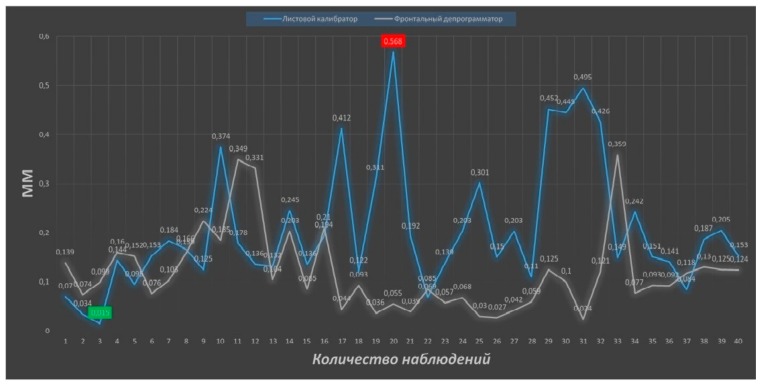
Figure of reproducibility graphic of leaf gauge (blue color) and frontal deprogrammer (grey color). Number of observations at the x-axis, value of deviation in mm at the y-axis.

**Table 1 ijerph-17-00933-t001:** Comparative assessment of the value of the difference in the mandible position. CR = centric relation.

The Determination Method of the CR	1 Bilateral Manipulation (n = 45)	2 Leaf Gauge (n = 45)	3 Frontal Deprogrammer (n = 45)	4 Gothic Arch Angle Recording (n = 45)	*p*
The average discrepancy of the mandible positions (in mm)	M ± m				p_1–3_ ≤ 0.05 p_1–4_ ≤ 0.05 p_2–3_ ≤ 0.05 p_2–4_ ≤ 0.05
	0.225 ± 0.028	0.207 ± 0.02	0.119 ± 0.012	0.120 ± 0.013	

**Table 2 ijerph-17-00933-t002:** Comparative analysis of occlusal contacts during CR determination in various ways.

Method of Determination CR The Value of the Discrepancy of the Occlusal Contact along the Axis (mm)	1 Frontal Deprogrammer (n = 45)	2 Gothic Arch Angle Recording (n = 45)	3 Leaf Gauge (n = 45)	4 Bilateral Manipulation (n = 45)	*p*
	M±m				
Vestibular–oral	0.039 ± 0.002	0.084 ± 0.004	0.464 ± 0.039	0.634 ± 0.04	p_1–3_ ≤ 0.05 p_1–__4_ ≤ 0.05 p_2–3_ ≤ 0.05 p_2–4_ ≤ 0.05
Mesiodistal	0.043 ± 0.002	0.054 ± 0.001	0.373 ± 0.04	0.388 ± 0.04	p_1–3_ ≤ 0.05 p_1–4_ ≤ 0.05 p_2–3_ ≤ 0.05 p_2–4_ ≤ 0.05

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
