# Peer review of "Comparative Analysis of the Reproduction Accuracy of Main Methods for Finding the Mandible Position in the Centric Relation Using Digital Research Method. Comparison between Analog-to-Digital and Digital Methods: A Preliminary Report"

_ijerph, 2020, doi:10.3390/ijerph17030933_

Round 1

Reviewer 1 Report

The present study was aimed to compare the aspect of the reproduction accuracy in studied methods of  determination of the CR of jaws using the digital research methods, that were the bilateral manipulation by Dawson P.E., the frontal deprogrammer, the leaf gauge and the intraoral device of recording the Gothic arch angle.

The topic of the manuscript is interesting in that field, but the article needs some improvements.

Title

I suggest identifying the present study as “pilot study” because the test was made on 5 subjects. Although the CR position was repeated for 20 registrations for each method, a more accurate evaluation should include a larger sample. 

Introduction

Please, improves and expands the part on the literature review which deals with the importance of the CR jaw position:

About the sentence: “Failures associated with the CR determination, with a high degree of probability can lead to occlusal and muscle disfunction and the appearance of pain not only in the craniomandibular system, but also in the craniocervical area”, please, also refer to “Surface electromyographic response of the neck muscles to maximal voluntary clenching of the teeth” by Ciuffolo et al.
In addition, also report about the correlations with low-back problems, as stated for example in the manuscript from Tecco et al. “SEMG activity of masticatory, neck, and trunk muscles during the treatment of scoliosis with functional braces. A longitudinal controlled study”.

It is necessary to add references in literature about the 5 methods used for determination of the CR jaw position.

In particular, the sentence: “Certain difficulties arise due to the significant existence of CR interpretations, indications and contraindications, as well as methods for its recording, some of which are very contradictory [3]. Nowadays, according to the foreign studies, there are more than thirty interpretations of the CR and there are many methods for its determination [9]” needs to be better explained.

Material and Methods

The present study compares 5 methods to assess the CR position, but also it compares 2 methods to evaluate the reproducibility (analog-to-digital and digital). This last question (the comparison between the analog-to-digital and the digital methods) must be included also in the Title of the manuscript.

Line 129: For the shell-shell comparison, please include a figure in which it is possible to see also areas with blue colors and red colors (areas of discrepancy)

Please, report data about a method error evaluation, regarding the procedure with photoshop (an intra-observed method evaluation, or inter-observers

Please specify how many operators made the elaboration of images

Results

Please, evaluate also the difference observed between the analog-to-digital and digital methods in terms of “correlation” between these two methods.

Discussion

Please discuss the difference between the analog-to-digital and digital methods, mostly to explain the following conclusion: “The Avantis 3D software identified the reproducibility of the mandible position the most in aspect of searching the position of centric relation”.

You state: “It is worth noting that in the comparative aspect we used analog-digital and digital methods to determine the precision revealed an identical trend of reproduction of the mandible position”, but this point needs to be accompanied by a comparison of data and a statistical analysis.

Tables

Please, improve table in their graphical design

Reviewer 2 Report

STUDY DESIGN

Due to the limited number of patients enrolled in this study I ask you to state in the article that is a “Preliminary report”. You should add “Preliminary report” also in the title.

INTRODUCTION

LINE 49 orthopedic rehabilitation?

Line 50- 52 “Failures associated with the  CR determination, with a high degree of probability can lead to occlusal and muscle disfunction and  the appearance of pain not only in the craniomandibular system, but also in the craniocervical area” According to a recent systematic review of the Literature there is a lack of evidence in associating Temporomandibular disorders (TMD) to dental occlusion. Please see Manfredini D. et al J Oral Rehabil 2017;44:908-923. You should change this statement according to the actual Literature.

Line 55 “according to the foreign studies” What is the meaning of this statement?

Line 69 “The individual plates are made of light-cured plastic for holding two of the 69 parts.” Please rephrase this statement.

METHODS

Please describe the population, you asserted that 5 patients were enrolled in this study. Please describe the sample i.e. number of male/females age and average age.

Please indicate wheter the patients attended a private practice or an Institution (i.e. University, Department…).

Please add the specific info of the product used (Company name, Nation…)

Line 119-120 “Digital prints were obtained using the intraoral scanner Trios, and using laboratory scanner Prime were obtained digital copies of the CR records, which are integrated into the computer program Avantis 3D.” Please rephrase this statement as the concept is not clear.

RESULTS

Line 168-170 “When analyzing the data by the analog-digital method, it can be noted that the smallest  discrepancy of the first contact position in the CR was observed when using the frontal deprogrammer (the discrepancy along the VO axis is 0.055±0.03 mm, along the MD axis is 0.047±0.02” Please just state the results living the comments to the Discussion section.

Round 2

Reviewer 1 Report

This manuscript improved a lot after a proper revision, and in my opinion, it can be published in the present form

thanks

Simona

Reviewer 2 Report

Dear Authors

you have satisfactory addressed all the review requests.

Best Regards